# Accurate Real Time On-Line Estimation of State-of-Health and Remaining Useful Life of Li ion Batteries

**Cher Ming Tan** [1,2,3,4,*], **Preetpal Singh** [1] and **Che Chen** [1]

1. Centre for Reliability Science and Technology, Chang Gung University, Wenhua 1st Road, Guishan Dist., Taoyuan City 33302, Taiwan; preetpalsingh96@gmail.com (P.S.); kobebrian787@gmail.com (C.C.)
2. Center for Reliability Engineering, Ming Chi University of Technology, New Taipei City 24301, Taiwan
3. Department of Electronic Engineering, Chang Gung University, Wenhua 1st Rd., Guishan Dist., Taoyuan City 33302, Taiwan
4. Department of Urology, Chang Gung Memorial Hospital, Guishan, Taoyuan City 33302, Taiwan
* Correspondence: cmtan@cgu.edu.tw; Tel.: +886-3-2118800-3872

**Abstract:** Inaccurate state-of-health (SoH) estimation of battery can lead to over-discharge as the actual depth of discharge will be deeper, or a more-than-necessary number of charges as the calculated SoC will be underestimated, depending on whether the inaccuracy in the maximum stored charge is over or under estimated. Both can lead to increased degradation of a battery. Inaccurate SoH can also lead to the continuous use of battery below 80% actual SoH that could lead to catastrophic failures. Therefore, an accurate and rapid on-line SoH estimation method for lithium ion batteries, under different operating conditions such as varying ambient temperatures and discharge rates, is important. This work develops a method for this purpose, and the method combines the electrochemistry-based electrical model and semi-empirical capacity fading model on a discharge curve of a lithium-ion battery for the estimation of its maximum stored charge capacity, and thus its state of health. The method developed produces a close form that relates SoH with the number of charge-discharge cycles as well as operating temperatures and currents, and its inverse application allows us to estimate the remaining useful life of lithium ion batteries (LiB) for a given SoH threshold level. The estimation time is less than 5 s as the combined model is a closed-form model, and hence it is suitable for real time and on-line applications.

**Keywords:** state of health; remaining useful life; electrochemistry based electrical model; semi-empirical capacity fading model; useful life distribution; quality and reliability assurance

---

## 1. Introduction

Electric vehicles (EV) are the focus of attention for today transportation, and their primary energy source is mainly rechargeable lithium ion batteries (LiB) due to their higher energy efficiency and longer lifetime as compared to their counterparts. However, the estimation of their health and the prediction of their remaining useful life (RUL) for EV are the major issues. In particular, the accuracy and rapid measurement of their health is a major concern.

The important health indexes for LiB have already been discussed extensively [1–5]. State-of-health (SoH), state of charge (SoC), state of energy (SoE), and state of safety (SoS) were discussed in detail for LiB. Inaccurate SoH estimation can lead to unintentional over-discharge as the actual $Q_m$ (the remaining charge in LiB) can be a lot lower and thus 20% SoC that are ready for re-charge could actually be much lower. Such inaccuracy or uncertainty in SoH estimation can also lead to being over conservative on

the user's part that increase the SoC cut off for battery charging, and lead to a higher number of charge cycles than necessary [6]. Both situations can accelerate the battery's degradation.

The unknown RUL of LiB could also result in conservative pre-mature replacement of LiB, increasing the cost of LiB in its applications [7–10]. On the other hand, if the SoH of LiB is actually already lower than 80%, prolonged usage of LiB could be problematic.

Accurate estimations of SoH and RUL are also crucial in energy storage applications. As the charging and discharging tend to be more often when LiB is used for energy storage, especially under solar and wind energy systems, large RUL with respect to charging and discharging will be important for a specific SoH threshold to justify the system cost. A sufficiently high SoH of LiB should be maintained for energy storage so that it can provide enough energy for usage when the solar energy or wind energy is no longer available. This model allows such estimation of the RUL.

Consequently, accurate estimation of SoH and prediction of RUL of LiB are crucial for LiB applications. It can be evaluated by comparing the present time performance with the ideal state performance and the battery's fresh state. There are many SoH estimation methods, and some are accurate but require complex calculation that is not suitable for real time applications. Some are not sufficiently accuracy for real applications.

The fading capacity and the increase in cell impedance of a LiB are two important factors that must be taken into consideration while estimating the SoH of the battery. The data-driven and adaptive systems are the two approaches that are generally implemented to determine the SoH of a battery. SoH estimation is performed using cycling data and parameters that affect the battery lifetime in data-driven approach. Battery's internal resistance is also used to determine SoH, but it is difficult to monitor internal resistance of the battery in the real-time scenarios. Also, deep understanding of correlation of battery operation and degradation process is required for this approach. Different designs of cells and operation conditions can have different degradation mechanisms as pointed out by Palacin [11], and hence such correlation can be difficult.

Adaptive systems approach makes use of the parameters that are sensitive to battery's degradation trend, and these parameters must be measured and examined throughout the battery operation time. However, high computational load complicates the online running of the model on a real application [12], and this is a huge drawback for adaptive systems.

Recently, Huang et al. [13] developed an online SoC and SoH estimation model for LiB. Their SoH estimation is obtained from the reciprocal of unit time voltage drop (V′), a parameter that was developed by them. However, the estimated SoH depends on the SoC at which the V′ is computed, and correction factor for the given SoC is to be obtained from a correction curve, which was done via curve fitting. This can render the model applicable only to the tested LiB as the underlying physics of the correction curve is not explored. Also, their SoH computation does not depend on the charge/discharge cycle, and hence it cannot be used for the estimation of the RUL.

As SoH is a measured of the remaining maximum charge that can be stored in LiB after several cycles of charge and discharge, and with the understanding of the degradation of LiB in term of its maximum stored charge [14], it is reasonable to assume that SoH is a function of the number of charge-discharge cycle, the discharge current, and temperature. Palacin [15] provided a good overview of the main ageing mechanism in LiB, and she showed that the combination of high current and high temperature are most detrimental to the degradation of LiB.

Liu et al. [16] proposed a semi-empirical capacity fading (SECF) model based on the above-mentioned factors, and Preetpal et al. [17] recently demonstrated the successful use of this model for the estimation of SoH of a set of LiB at different charge-discharge cycles, and the estimation error was less than 2.5% when the extrapolation went beyond 300 cycles of operation. However, their work did not consider the effect of temperature and discharge rate in the estimation of SoH. In this work, we extend our investigation of the model in estimating SoH of LiB under different temperatures and discharge rates. As the equation is in a compact closed form, it can be implemented easily in real time, and we could also able to use the equation to determine the cycle at a specific SoH threshold, such as 80%, making prediction of its RUL possible in

both EV and energy storage applications. A noteworthy point is that the depth of discharge of LiB during operation is not considered in this work although it is known to affect the SoH of LiB. This is because of its complex degradation mechanism for LiB, and it will be treated in the future work.

To evaluate the accuracy of the SoH estimation using the SECF model, we use the electrochemistry-based electrical model (ECBE) model [18] as reference. The ECBE model is based on the first principle of electrochemistry, which is unlike the previously reported models, which employ the best-fit techniques. ECBE model results are also verified using the electrochemical impedance spectrometer (EIS) results. In fact, EIS is also used to understand the various aging mechanisms using electrical models, but the method can be done only offline in frequency domain. On the other hand, ECBE allows the performance of each component inside LiB be determined real time through its discharging curve non-destructively (i.e., terminal voltage vs. time during discharge), making it suitable for field applications. Although this ECBE model is accurate, the computation time to obtain the values of the model parameters is too excessive, which renders its unsuitability for in-situ real time SoH estimation.

The paper is organized as follows. The experimental settings and approach are given in Section 2, and a brief introduction of the semi-empirical capacity fading model is shown in the subsequent section. The applications of the SECF model for cases of different discharge rate and temperature as well as its SoH estimation accuracy are presented next. Conclusions and future works are given in the last section.

## 2. Materials and Methods

4 Panasonic SANYO UR18650E lithium-ion batteries [19] are tested using Bio-Logic BCS-815, which is an 8-channel tester and support 15 A per channel. It is used for charging and discharging of the batteries with test data collection under room temperature, which is approximately 25 °C. The sampling frequency of the data collection is 1 Hz. The specification of the batteries used here are shown in Table 1 as obtained from the manufacturer.

**Table 1.** Samsung 18,650 battery specification provided by manufacturer [8].

| Battery Characteristics | |
|---|---|
| Type | Cylindrical |
| Chemical system | NMC |
| Nominal voltage | 3.62 V |
| Typical capacity | 2150 mAh |
| Cut-off voltage Charging | 4.2 V |
| Discharging | 2.75 V |
| Dimensions(mm) | $18.4 \times 65$ |
| Approx. weight | 44.5 g |

The batteries are tested under different test conditions as shown in Table 2. Only one battery is tested under each condition due to time and resources limitations. However, as the model employed here has been verified in the work by Preetpal et al. [17] on different charge-discharge cycles for LiB, the results obtained in this work are presented with confidence despite the small sample size. Verification of the model using another 3 cells from the same batch will be performed later. The purpose of this work is to examine the capability and accuracy of this SECF model when ambient temperature and discharge current are considered. More batteries will be tested when resource becomes available.

**Table 2.** Test conditions for the batteries.

| Cell Name | Test | Temperature (°C) | Discharge Current |
|-----------|------|------------------|-------------------|
| A | 1 | 55 | 1 C |
| B | 2 | 55 | 3 C |
| C | 3 | 25 | 1 C |
| D | 4 | 25 | 3 C |

In the experiments, we use CC-CV method for charging the batteries. 1 A constant current (as recommend from the battery's manufacturer datasheet) is used to charge the batteries to its cut-off voltage, and the charging is done with constant voltage until the current drops to 100 mA. When the battery is fully charged, a rest time of 30 min is kept before discharging. One or 3 C-rate is implemented to discharge the batteries until the cut-off voltage of 4.2 V is reached. Figure 1 shows the typical time progression of the terminal voltage of a LiB during charging and discharging. One can see that the voltage is higher when LiB is charging at higher temperature. This is because cell impedance will be higher at higher temperature. The terminal voltage can be modelled as the constant current multiples the cell impedance as the charging is done using CC-CV, and hence a higher terminal voltage results. In fact, the cell impedance of a LiB has been employed to monitor the temperature of LiB by Beelen et al. [20].

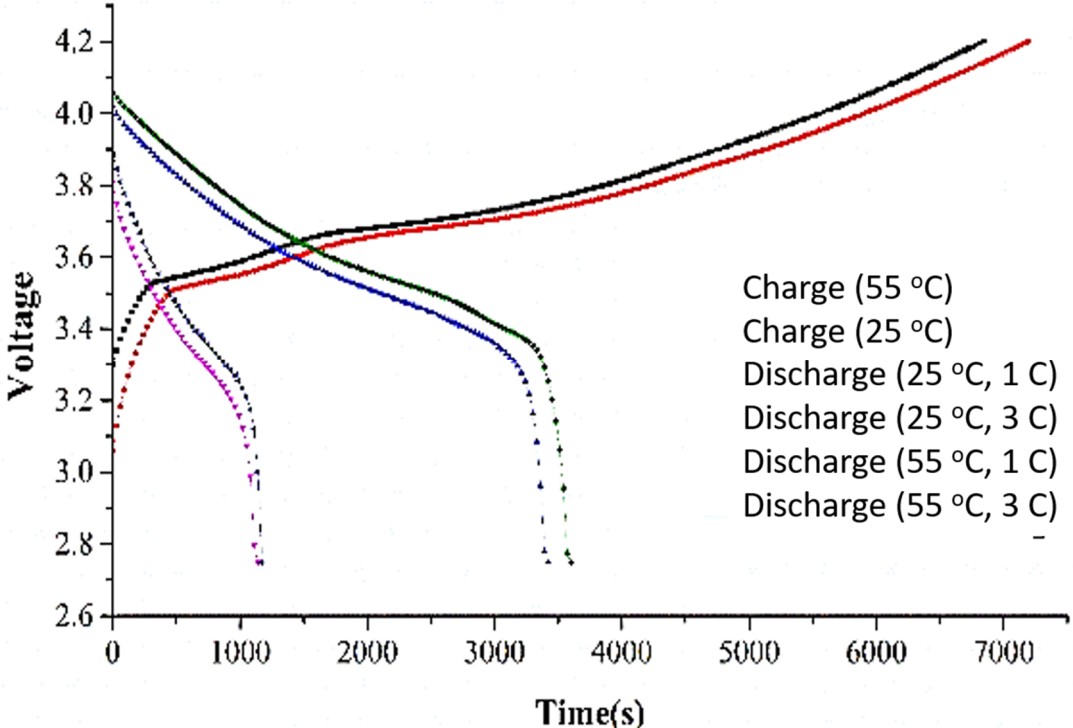

**Figure 1.** Variation of terminal voltage recorded during discharge period for 1 and 3 C-rate at different temperature.

The discharge curves of all the LiB are analyzed using the ECBE model to calculate the maximum capacity of the cells ($Q_m$). This $Q_m$ is used to compute the corresponding SOH using Equation (1) where $Q_{max(fresh)}$ is the $Q_m$ after the first discharge cycle, and $Q_{max(aged)}$ is the $Q_m$ after the subsequent cycles. The SoH computed is termed as Experimental SoH when the $Q_m$s are determined using ECBE model.

$$\text{SoH} = \frac{Q_{max(aged)}}{Q_{max(fresh)}}$$

(1)

The SECF model used in this work is shown in Equation (2) [16]. The SoH determined using Equation (2) is termed as Estimated SoH. The parameter $k_1$ accounts for the capacity losses that increase rapidly during the conditions of cycling at high temperature, and $k_2$ is a factor to account for capacity losses under the normal conditions of cycling. $k_3$ accounts for the capacity loss due to C-rate [21].

$$SoH = 1 - \left(0.5*k_1N^2 + k_2N\right) - \frac{k_3}{Q_{max(fresh)}}i \tag{2}$$

Here, N represents the number of charge-discharge cycles the battery experienced at the time of this SoH calculation and i is discharging current.

These parameters ($k_1$, $k_2$, and $k_3$) can be extracted by substituting the Experimental SoH obtained from Equation (1) into Equation (2) at 3 different cycles (the exact cycles can be seen in Table 3 later in order to ensure the largest cycle chosen among the 3 cycles is half of the cycle at which the SoH of the LiB is around 80%) as shown in Figure 2. The extracted parameters are then used to calculate the Estimated SoH in this work. Our experimental procedure is depicted in Figure 3. There are large fluctuations or non-linearity observed in Qm over the first few cycles, and this non-linearity is higher at lower temperatures, which is also observed by other researchers [14], thus the first 50 cycles are not used for the k values extraction.

**Table 3.** k values of the semi-empirical capacity fading (SECF) model obtained at four test conditions.

| Temperature (°C) | C-Rate | 3 Cycles for the Extraction of k Values | $k_1$ | $k_2$ | $k_3$ |
|---|---|---|---|---|---|
| 25 | 1 | 100, 200, 300 | 0 | 0.000283 | 0.0027 |
| 25 | 3 | 100, 200, 300 | 0 | 0.0000599 | 0.0101 |
| 55 | 1 | 100, 200, 300 | 0 | 0.000354 | 0 |
| 55 | 3 | 75, 125, 250 | 0 | 0.00045 | $1.43 \times 10^{-3}$ |

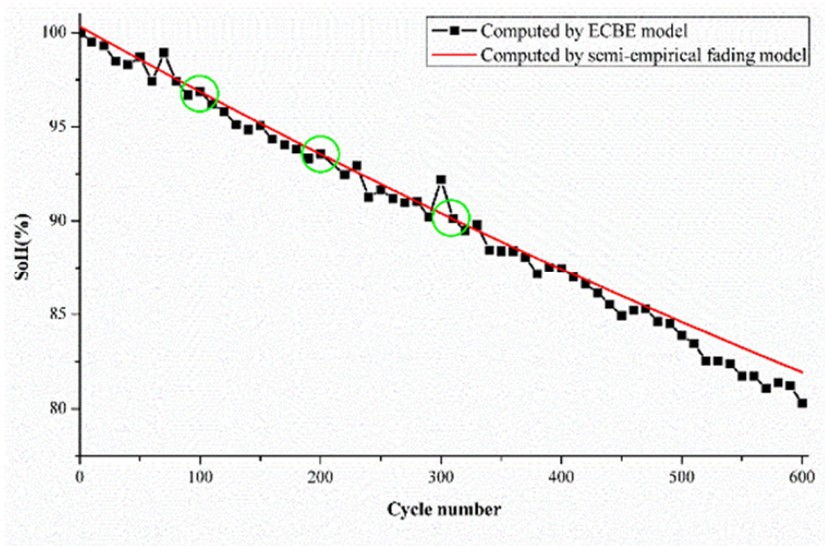

**Figure 2.** State-of-health (SoH) estimation results from ECBE model (black curve) and semi-empirical fading model (red curve) for battery tested at 55 °C and 1 C discharge current. Green circle represents the points used to find the k values.

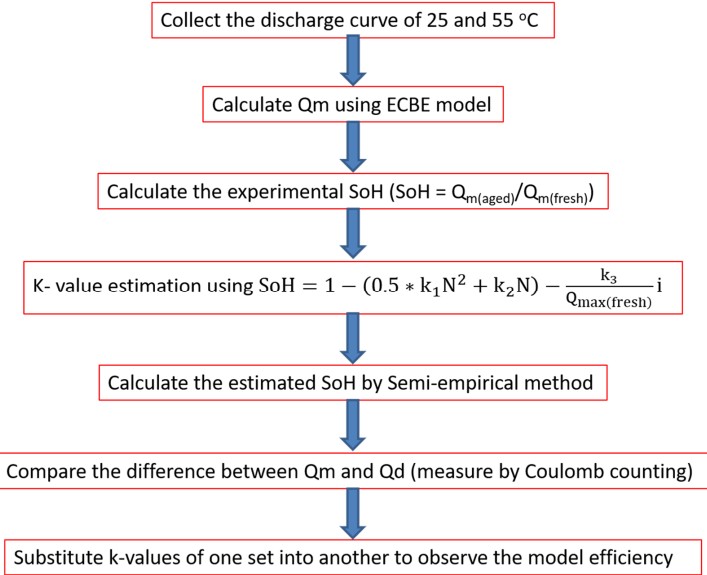

**Figure 3.** Flowchart for error estimation between experimental and estimated SoH.

## 3. Results

To demonstrate the effect of varying C-rate at constant temperature and the effect of varying temperatures at a constant C- rate on SoH estimation accuracy of the SECF model, we divide our results into four sections as shown in Figure 4. In Figure 4, A, B, C and D represent cases where k values are obtained from batteries tested at 1 C_55 °C, 3 C_55 °C, 1 C_25 °C, and 3 C_55 °C, respectively. On the other hand, A′ and B′ represent the SoH estimation errors for cells A and B using k values obtained from cell A. C* and D* represent the SoH estimation errors for cells C and D using k values obtained from cell A. Similarly, A″ and B″ are the SoH estimation errors for cells A and B using k values obtained from cell B. Others follow the same notations.

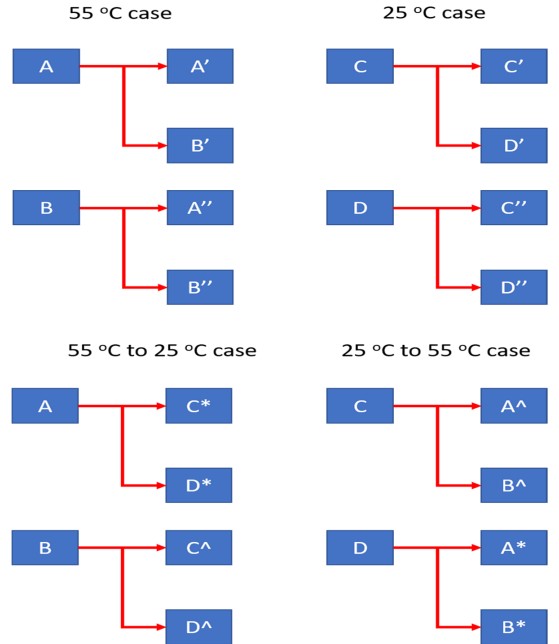

**Figure 4.** Notations used in the analysis of test results.

*3.1. Effect of C-Rate at Constant Temperature Conditions*

3.1.1. SOH Estimation for Batteries under Different Discharge C-Rate at 55 °C

Experimental SoH is calculated for each cell tested for different number of cycles at two C-rates and 55 °C, and the results are shown in Table 4. SoH decreases with the cycle number and the rate of decrease of SoH is higher for battery discharged at 3 C rate as expected. The sample with 1 C-rate reaches the cut-off point of around 80% SoH after 600 cycles, while the sample with 3 C-rate reaches the cut-off point before 500 cycles.

**Table 4.** Experimental SoH at 55 °C based on the Qm computed from the ECBE model.

| Cycle Number | 1 C | 3 C |
| --- | --- | --- |
| 0 | 100 | 100 |
| 100 | 96.86 | 95.24 |
| 200 | 93.55 | 91.29 |
| 300 | 90.21 | 87.68 |
| 400 | 87.46 | 82.98 |
| 500 | 83.88 | 79.67 |
| 600 | 80.28 | - |

The calculated SoH values shown in Table 4 are used to obtain the k's values of the SECF model, and the values are shown in Table 5. The very small negative values of $k_1$ is likely due to fitting approximation error and hence they are set to zero. In fact, $k_1$ value should be zero as our ambient temperature is not high as expected from the work by [21].

**Table 5.** The k-values of 1 and 3 C-rate, respectively, at 55 °C.

| | $k_1$ | $k_2$ | $k_3$ |
| --- | --- | --- | --- |
| 1 C-rate of 55 °C (A) | $-1.61478 \times 10^{-7}$ | $3.5497 \times 10^{-4}$ | 0 |
| 3 C-rate of 55 °C (B) | $-3.7322 \times 10^{-7}$ | $4.5086 \times 10^{-4}$ | $1.43376 \times 10^{-3}$ |

To verify the accuracy of the SECF model, the k-values obtained from one battery is applied to obtain the SoH of same LiB. The percentage error is then evaluated by comparing the estimated SoH with the experimental SoH. Figure 2 shows such comparison and the result is satisfactory.

To study the effect of C-rate on the estimation accuracy, the k-values obtained from one battery is applied to obtain the SoH of other battery tested under same temperature but at other C-rate. Figure 5 summarizes the percentage errors at other discharge conditions. It is to be noted that the maximum percentage error on estimating SoH using semi-empirical model is around 3.14% using the k-values of battery tested at 1 C and applied to battery tested at 3 C, and the percentage error is around 2.11% for the reverse case. Both errors are within the acceptable limit of 10%.

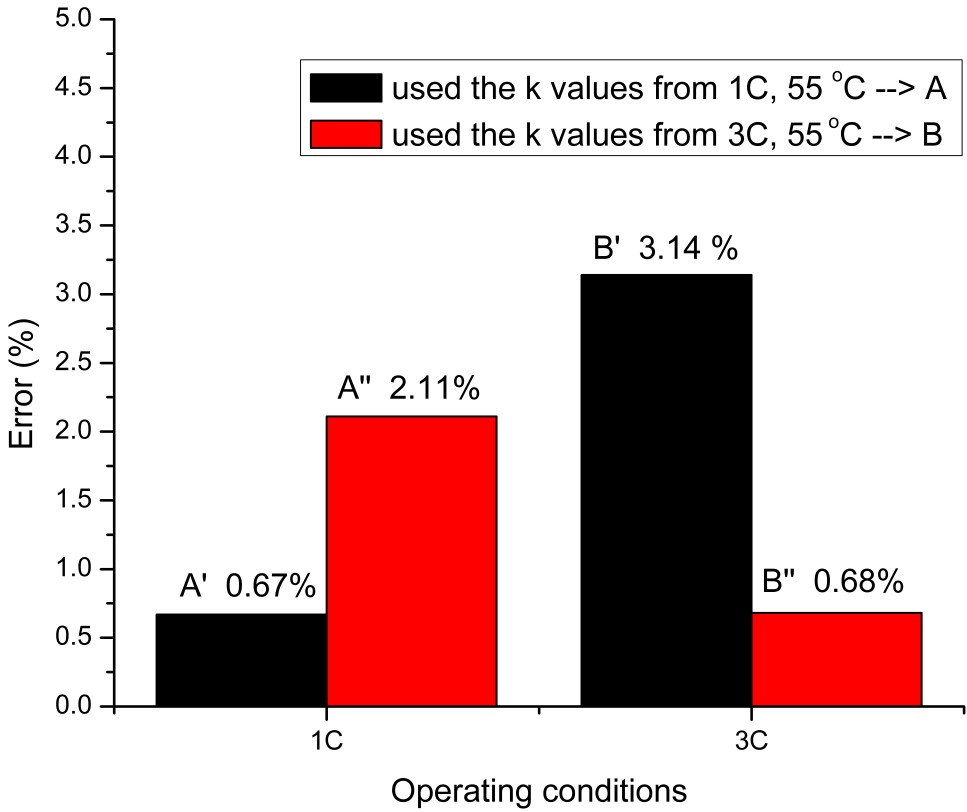

**Figure 5.** The estimation error between different discharge current under 55 °C. % value on top of the bar represents the % error in estimation.

From Figure 5, we can see that the estimation error of using the SECF model is sufficiently accurate for predicting the SoH of its own cell at larger cycle number, and this has been elaborated in the work by Preetpal et al. [17]. We can also see that it is fairly accurate when it is applied to the cells at different C rates. In comparing the percentage errors of A″ and B′, although the different in the C rate is the same, the largest percentage error is observed for the case of B′. This is expected because of the largest temperature change of the cells from the reference cell to the cell of interest in the case of B′. We will discuss this in more detail later. Hence, we see that both the different in the discharge rate and the cell temperature can affect the estimation accuracy.

In terms of the selection of the cycles to estimate the k's values, we select a different set of the three cycles for the SoH estimation of battery tested at 55 °C and 1 C rate. Table 6 shows the comparison, and one can see that the effect of the selection on the percentage error in SoH estimation is not significant.

**Table 6.** Percentage estimation error of SoH for a different set of three cycles in the k's values extraction.

| Cycles Selected for k's Values Extraction | % Estimation Error |
| --- | --- |
| 100, 200, 300 | 3.14 |
| 125, 225, 325 | 3.68 |
| 150, 250, 350 | 3.52 |

### 3.1.2. SoH Estimation for Batteries at Different C-Rates under 25 °C Ambient

We perform a similar study as in the previous section to estimate SoH for batteries tested at two C-rates and 25 °C, and the results are shown in Tables 7 and 8. Batteries tested under 25 °C shows lower degradation rate at both 1 C-rate and 3 C-rate when compared with batteries tested at 55 °C as

expected. Also, the degradation rate is slower for 1 C-rate as expected. Sample using 1 C-rate still have 89.76% SoH after 800 cycles, while the 3 C-rate sample reaches the cut-off point at around 700 cycles.

**Table 7.** Calculated SoH at 25 °C based on the Qm computed from the ECBE model.

| Cycle Number | 1 C | 3 C |
|:---:|:---:|:---:|
| 1 | 100% | 100% |
| 100 | 97.38% | 96.28% |
| 200 | 94.91% | 93.00% |
| 300 | 94.26% | 91.30% |
| 400 | 93.14% | 88.21% |
| 500 | 92.48% | 84.48% |
| 600 | 91.88% | 83.80% |
| 700 | 90.81% | 80.41% |
| 800 | 89.76% | - |

**Table 8.** The k-values of 1 and 3 C-rate respectively at 25 °C. Negative value of $k_1$ is set to zero.

| k-Values | $k_1$ | $k_2$ | $k_3$ |
|:---:|:---:|:---:|:---:|
| 1 C-rate of 25 °C (C) | $-8.9785 \times 10^{-8}$ | $2.8312 \times 10^{-4}$ | 0.0027 |
| 3 C-rate of 25 °C (D) | $8.3792 \times 10^{-7}$ | $5.9967 \times 10^{-5}$ | 0.0101 |

Again, the k's values obtained from one battery is applied to obtain the SoH of same battery as well as other battery tested at other C-rate and the error percentage is computed between the Experimental and Estimated SoH. The percentage error results for estimating SoH are shown in Figure 6.

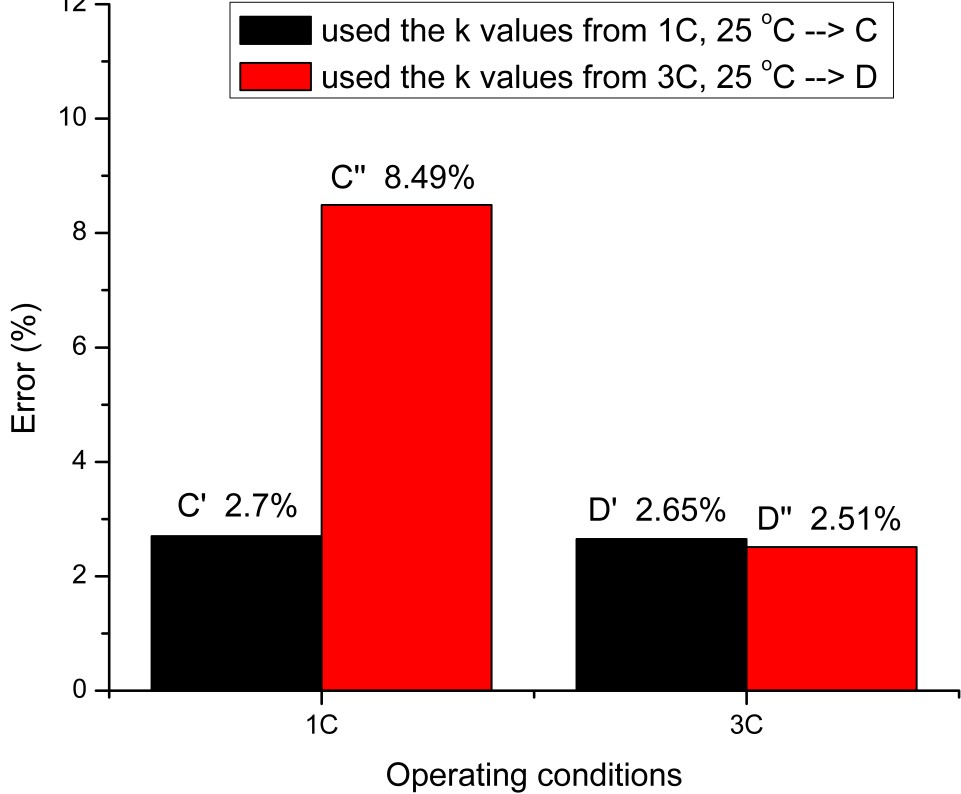

**Figure 6.** The estimation error between 1 and 3 C-rate at 25 °C. The % value on top of each bar represent the % error of estimation.

In comparing the % errors for C′ and A′, and that for D′ and B′, we can see that the % error is larger for the case of 25 °C. Such larger error is due to the inaccuracy of the temperature of 25 °C as the testing was done in the laboratory ambient, and the ambient temperature has a small fluctuation over the span of measurement period. The temperature varies between 24 to 26 °C during our test.

In Figure 6, the largest percentage error obtained is around 8.49 % for the case when k's values from battery tested at 3 C is applied to obtain SoH for battery tested at 1 C under 25 °C. In comparison to D′, although the difference in the C rate between the cells are the same, the large % error for the case of C″ is expected to be attributed by the temperature difference between the cell of interest and the reference cell, which is approximately 8 °C as can be seen in Table 9. Table 9 shows the temperatures of the LiB when discharged at different C rates at different temperatures.

**Table 9.** Cell temperature when discharged at different C rate under two ambient conditions.

|  | 1 C Rate | 3 C Rate |
| --- | --- | --- |
| 25 °C Ambient | 34.86 °C | 42.32 °C |
| 55 °C Ambient | 58 °C | 63 °C |

The increase in the cell temperature as shown in Table 9 is expected according to the work by Huang et al. [22], and the rise in temperature during discharge depends on the Peltier heat as a result of the entropy change, the discharge current, heat capacity and weight of a battery as well as its thermal design and the ambient temperature [22]. As the different C rate and cell temperature can affect the estimation accuracy, we further investigate the effect of temperature on the accuracy of the estimation using the SECF model.

### 3.2. Effect of Temperature

#### 3.2.1. SOH Estimation for Batteries at 25 °C using 55 °C Battery's Parameter Values (55 °C to 25 °C Case)

Similar to our previous approach, we use the k values from the batteries with similar C-rate but apply them to SoH estimation of cells discharged at different temperatures. First, SoH estimation is performed for batteries tested at 25 °C and the k values obtained from battery tested at 55 °C are used for the SoH estimation, and the results are shown in Figure 6.

It can be observed from Figure 7 that the highest SoH estimation percentage error of 8.4% is obtained for the case when k values of battery tested at 3 C rate and 55 °C is used to estimate the SoH for battery tested at 1 C rate and 25 °C. This is due to the largest cell temperature difference between the cells, which is approximately 28.14 °C in this case as can be seen in Table 9. The smallest estimation error for the estimation of battery tested at 3 C and 25 °C is corresponding to the small difference in cell temperature for cell discharge at 1 C and 55 °C and cell discharge at 3 C 25 °C. This finding supports our postulation that cell temperature difference is an important factor in the estimation accuracy.

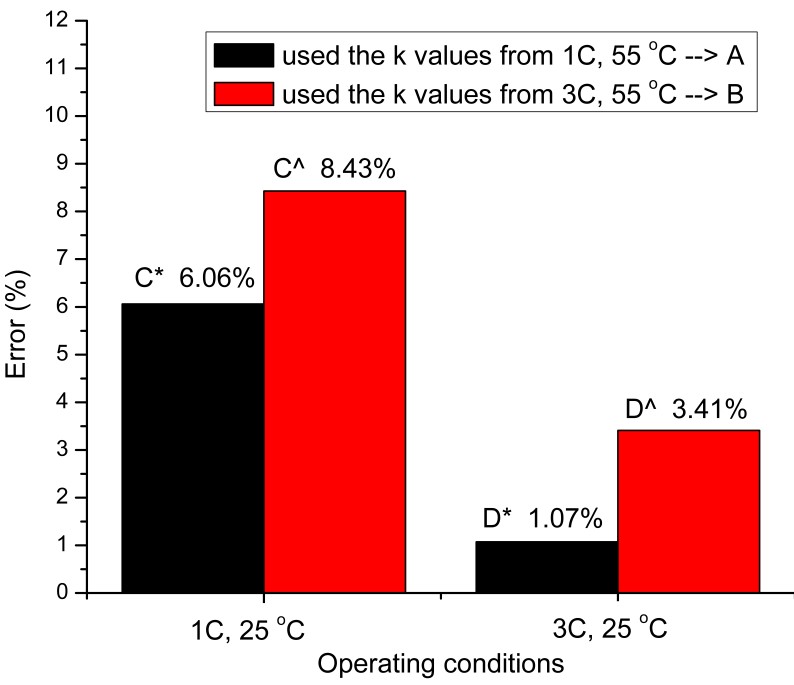

**Figure 7.** Estimate % error using k's values from 55 °C.

3.2.2. SOH Estimation for Batteries at 55 °C using 25 °C Battery's Parameter Values (25 °C to 55 °C Case)

Now we study the opposite situation where estimation is made using parameters values from low temperature to high temperature, and the results are shown in Figure 8.

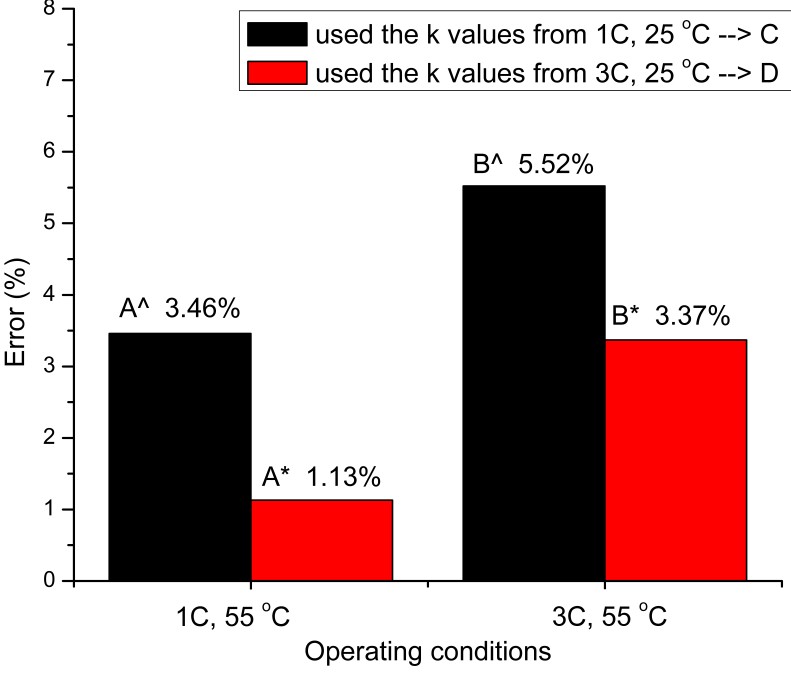

**Figure 8.** Estimation % error using k values from 25 °C.

From Figure 8, we can see that the smallest percentage error is the case where the parameters values from 3 C and 25 °C is applied to 1 C at 55 °C. This is expected with our postulation where the cell temperature difference between the two cells is only 15.68 °C in this case. Likewise, the largest

percentage error is observed for the case where the parameters values from 1 C at 25 °C is applied to 3 C at 55 °C and the temperature difference is as high as 28.14 °C.

From the above investigation, the interaction of discharge current and cell temperature needs to be identified. The statistical design of experiment (DoE) method, a commonly used method in industry for studying the effect of multiple factors, is thus employed. The detail of the DoE method can be found in reference [23]. We summarized the k values at the four test conditions as shown in Table 3. The interaction of the discharge current and temperature as reflected in the k values can be plotted as shown in Figures 9 and 10, as part of the DoE analysis. Using these values and applying the $2^2$ factorial design analysis, we obtain the equations for $k_2$ and $k_3$ as given in Equations (3) and (4). $k_1$ is zero as our experiments do not involve high temperatures.

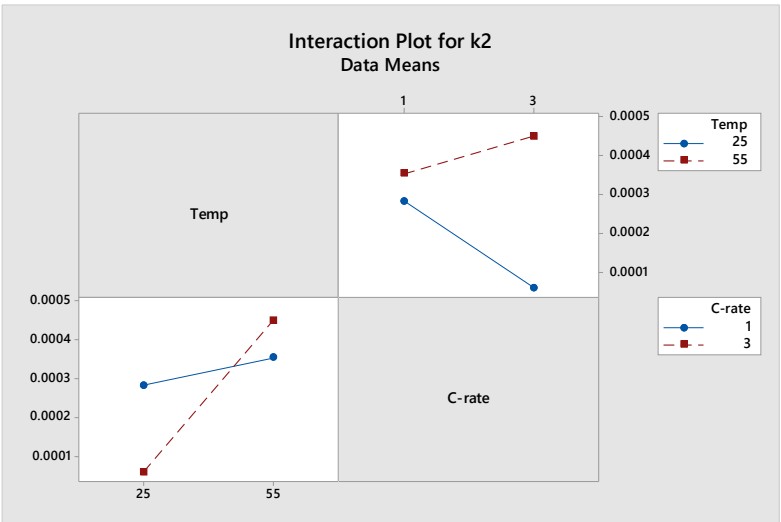

**Figure 9.** Interaction of current density and temperature in the $k_2$ value.

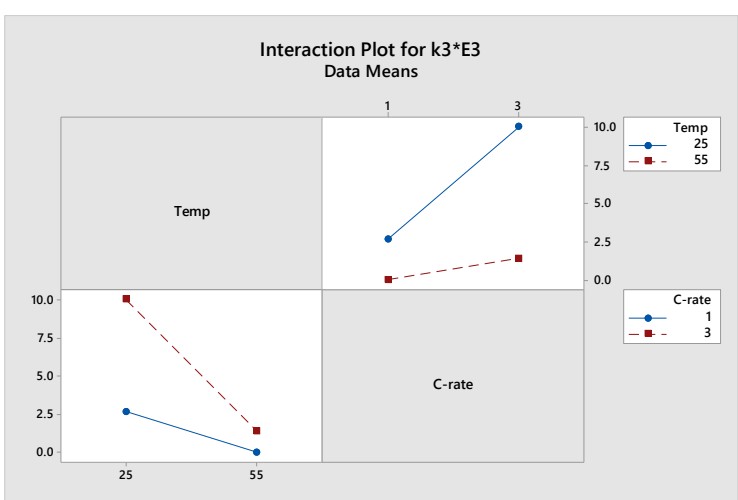

**Figure 10.** Interaction of current density and temperature in the $k_3$ value.

From the DoE analysis, we obtain the following equations for $k_2$ and $k_3$.

$$k_2 = 0.000287 - 0.000115\,A - 0.000080\,B - 0.000032\,A \times B \tag{3}$$

$$k_3 = 0.003557 + 0.002207\,A + 0.002843\,B + 0.001493\,A \times B \tag{4}$$

where

$$A = (\text{Temperature (°C)} - 40\,°C)/15\,°C \tag{5}$$

and

$$B = C\ \text{rate} - 2 \tag{6}$$

Using Equations (3) and (4), we can estimate the SoH via Equation (2), and the percentage errors in the SoH estimation with respect to the Experimental SoH can be seen in Table 10. One can see that with the inclusion of the interaction of discharge current and temperature, the % errors in the SoH estimation are consistently less than 1%.

**Table 10.** Comparison of the % errors in SoH estimation.

| Test Condition | SoH (Using ECBE) | % Error in SoH Estimation Using Its Own k Values | % Error in SoH Estimation with k Values Computed from Equations (3) and (4) |
|---|---|---|---|
| 25 °C_1 C | 89.76(after 800 cycles) | 2.7 | 0.89 |
| 25 °C_3 C | 80.41(after 300 cycles) | 2.51 | 0.77 |
| 55 °C_1 C | 80.29(after 600 cycles) | 0.67 | 0.59 |
| 55 °C_3 C | 79.68(after 500 cycles) | 0.68 | 0.41 |

We also extend our equations to the cases of 0.5 C and 5 C at 55 °C. The % errors in the estimation are shown in Table 11, and one see that the % error is small even for the case of 5 C where cell heating is serious. As 5C at 55 °C is likely to be the highest cell temperature for LiB, this implies that $k_1$ is always zero for the case of LiB

**Table 11.** % error in SoH estimation for the cases of 0.5 and 5 C at 55 °C

| Test Condition | SoH (Using ECBE) | % Error in SoH Estimation | % Error in SoH Estimation with Generalized k Values |
|---|---|---|---|
| 55 °C_0.5 C | 80.27(After 600 cycles) | 0.97 | 0.84 |
| 55 °C_5 C | 76.08(After 200 cycles) | 5.47 | 0.61 |

To verify the proposed model, and for the purpose of reproducibility, three other batteries from the same batch are tested at 55 °C_5 C and Table 12 shows the % errors in the SoH estimation estimated using Equations (3) and (4). The test condition is chosen to save time as the battery degradation is fastest in such harsh condition. Batteries are tested for 100 cycles. The results in Table 12 shows the good accuracy of the proposed method.

**Table 12.** Verification of the model for additional three more batteries tested at 55 °C and 5 C discharging current.

| Sample # | SoH (Using ECBE) | SoH Estimation Using Our Generalized k Values | % Error in SoH Estimation with Generalized k Values |
|---|---|---|---|
| 1 | 89.06(After 100 cycles) | 88.24 | 0.92 |
| 2 | 91.39(After 100 cycles) | 90.72 | 0.73 |
| 3 | 90.66(After 100 cycles) | 89.94 | 0.79 |

## 4. Applications of the SECF Model

From the previous analysis, we can see that with the equation of SoH given by Equation (2) and the k values computed using Equations (3) and (4), one can estimate the SoH of LiB after different charge-discharge cycle. Conversely, one can also determine the cycles where the SoH will reach a certain value due to its closed form. For example, if the LiB cell is operating under 2 C discharge rate at 40 °C, one can have the following equation, with i = 2C, $k_2$ = 0.00028, $k_3$ = 0.003557,

and $Q_{max(fresh)}$ = 1.9463 Ah. Upon solving it, the *n* value is 689, which will be the cycles where the SoH reaches 80%, and thus the useful life of the LiB cell at different operating condition can be determined.

$$80 = 1 - (k_2 N) - \frac{k_3}{Q_{max(fresh)}} i \qquad (7)$$

The above example is an application of the SECF model for the lifetime prediction of a fresh LiB. This model can also be used for the estimation of the RUL as follows.

Given the $k_2$ and $k_3$ values of a LiB as in this work, and assuming the LiB is used in 1 C discharge rate and has undergone 100 cycles at 30 °C, the LiB is now to be used in 1.5 C discharge rate at the same temperature of 30 °C. If this LiB has been through 20 cycles under 1.5 C discharge rate, we can use our SECF model to compute the RUL of the LiB under this 1.5 C discharge rate as follows.

For the SoH of the LiB after the 100 cycles @1C discharge rate, A = −10/15 and B = −1 according to Equations (5) and (6), and with the values of $k_2$ and $k_3$ as computed using Equations (3) and (4), we obtain the SoH after 100 cycles to be 96.25%, using Equation (2). This is the SoH of the LiB at the start of the next application of 1.5 C. The equivalent cycle for the LiB to reach this SoH at 1.5 C and 30 °C can be computed from the model by having the B = −0.5. A will not change as the temperature remains the same. With this new B, we have new values of $k_2$ and $k_3$ and the equivalent cycle is found to be 88 cycles ($N_{equivalent}$).

We can also compute the total number of cycles at 1.5 C and 30 °C to reach 80% SoH, denoted as $N_{total}$. From our SECF model, the value is found to be 769 cycles. Since the LiB has been through 20 cycles under 1.5 C, the RUL of the LiB will be $N_{total} - N_{equivalent} - 20 = 769 - 88 - 20 = 661$ cycles.

Going forward, if the discharge current varies, one can use the average current, expressed in terms of C rate to compute the B value. If the temperature is varying, one can use the average temperature to compute the A value. Then, using Equations (2)–(4), the SoH or the RUL can be determined. However, this is to be verified as our future work.

## 5. Future Works

While the SECF model is promising as shown in this work, the sample size used in this work is too small. In order to increase the confidence of this model, larger sample size and with LiB from different manufacturers as well as different type of LiB are to be used. This requires large resources and long test time. Collaboration with other research organizations on LiB is desired.

As mentioned in the introduction, the proposed model does not include the effect of the depth of discharge, which is known to affect SoH. This is one of the limitations of the proposed model and the inclusion of the depth of discharge in the model will also be our future work. Another limitation is the possible change of the values of the k parameters when the SoH becomes low as the internal structure of the LiB could have changed so significantly that a new set of k parameters are to be extracted from the ECBE model which has shown to be accurate even at low SoH [14]. Such investigation will also be our future work.

Besides the above-mentioned applications, other applications of SECF model are possible and yet to be verified. One of them is the development of acceleration model of LiB lifetime. If we can perform the same four set of test conditions at higher temperature and discharge rate, the k values can be obtained in a short time, and hence, the complete SoH estimation equations can be obtained and applied to compute the lifetime of LiB cell in other operating conditions, in the same way as in the above examples, for a given heat capacity, weight, and thermal design of the cells. Thus, battery manufacturers can simply provide a lifetime model for each batch of their batteries for their users. Also, as k values determined the lifetime of LiB, statistical distribution of the k values from a set of LiB samples in a production batch can also provide information on the lifetime distribution of the batch of LiB, and hence a process control methodology can be developed.

## 6. Conclusions

The requirement of accurate online estimation of LiB battery capacity and its remaining useful life (RUL) are important. In this work, we applied ECBE model and semi-empirical capacity fading (SECF) model to estimate SoH and its remaining useful life after a LiB is operated for different charge-discharge cycles. ECBE model is only used for initial SoH calculation twice to determine the parameters in the SECF model, which can then be used for the SoH and RUL estimation of cells for their subsequent charge-discharge cycles.

We demonstrate that SECF model for SoH estimation can provide accurate SoH estimation, using the ECBE model as reference, regardless of the discharge current and ambient temperature. The discharge rate considered is from 0.5 C to 5 C, and the ambient temperature is from 25 °C to 55 °C. As the model is in compact closed form, it can be used for the lifetime prediction of LiB, and this could provide a good method for LiB acceleration test so that the SECF model parameters can be determined for its lifetime estimation of LiB in other operating conditions in a short time. Also, with this closed form, this model is practical for on-line real time SoH estimation, as the estimation time after each discharge cycle is less than 5 s using a personal computer with capability of 8 GB RAM and Intel Core i5 processor.

**Author Contributions:** C.M.T. conceived and designed the research; P.S., C.C., and C.M.T. analyzed the testing data; P.S., C.C., and C.M.T. wrote the paper. All authors have read and agreed to the published version of the manuscript.

**Funding:** This research was funded by Chang Gung University and Ming Chi University of Technology, Taiwan and The APC was funded by Chang Gung University, Taiwan.

**Acknowledgments:** The authors would like to acknowledge the support of Centre of Reliability Science and Technologies Lab, Chang Gung University and Center on Reliability Engineering, Ming Chi University of Technology for providing valuable equipment and support for smooth conduct of experiments.

**Conflicts of Interest:** The authors declare no conflict of interest.

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
