# Peer review of "Accurate Real Time On-Line Estimation of State-of-Health and Remaining Useful Life of Li ion Batteries"

_applsci, doi:10.3390/app10217836_

Round 1

Reviewer 1 Report

Dear Authors,

First of all, my sincere thanks for your manuscript.

I have read your manuscript very carefully and I am pleased to discuss the following:

The entire text needs to be edited by a native speaker as I have found numerous English grammar errors. Too many to discuss here. 

Because I want to deal now more with the content, I strongly advice you to have your text be corrected by a native English speaker.

Now about the content:

Line 112-113: This brings me directly to a great dilemma: this paper cannot be accepted in the present form as we need reproducibility by testing at least 3 identical batteries for each condition. 

Line 115: Reproducibility is what you measure yourself! It is about your paper, not what others do.

Line 117: more batteries will be tested when resource is available. Why did you sent then this manuscript since you understand perfectly well you need reproducibility by testing more than one battery for each condition. This statement in Line 117 is actually why this paper cannot be accepted at this stage. 

Figure 1: Why is charge voltage higher at 55 degrees Celsius?

Figure 3: All letters are not well readable! Please make this Figure again.

Figure 4 is not clear to me. Explain again.

General: use in the text k1 and not k1, use k2 and not k2, use k3 and not k3.

Line 234-235: What is temp range lab? Why were tests not performed in a thermostatted oil bath?? Now the temperature is not defined and that is not acceptable.

Line 350-351: You conclude yourself that data are too limited so it is no surprise I conclude the same. 

I trust upon your expertise to:

  • run more tests with at least 3 batteries for one condition
  • stabilize the temperature in a temperature regulated oil bath
  • explain in more detail the model with k1, k2 and k3 as parameters
  • have the entire text be corrected by a native English speaker
  • strongly improve the Figures and graphics to make them more readable

I will advice the Editor the following:

The topic is very interesting but the revision needed will take more time than acceptable. Hopefully you will be able to find financial resource.

This means the peer review process end here for this manuscript as you first need to run more tests. I am very pleased to review your renewed manuscript. I wish you success in performing the additional tests and to take care to of the points as mentioned by me. 

As you see, the topic is very interesting but we simply need more and better defined tests.

Kind regards,

Reviewer

Author Response

Dear Editor,

We appreciate very much the effort of your team and the comments from Reviewers.  They are very good and have indeed help to improve our manuscript significantly. We took every comments seriously, and below are our replies for your considerations.  Thank you

We have also attached the manuscript files with highlights.

Reviewer 1

Dear Authors,

First of all, my sincere thanks for your manuscript.

I have read your manuscript very carefully and I am pleased to discuss the following:

The entire text needs to be edited by a native speaker as I have found numerous English grammar errors. Too many to discuss here. 

Because I want to deal now more with the content, I strongly advice you to have your text be corrected by a native English speaker.

Now about the content:

Line 112-113: This brings me directly to a great dilemma: this paper cannot be accepted in the present form as we need reproducibility by testing at least 3 identical batteries for each condition. 

Answer:

We fully agree with you, and we found another 3 batteries tested at 55oC@5C condition as shown below. Using our generalized k’s values as obtained, we found that the accuracies in SoH estimation are good.  This increase our confidence on the method proposed.  We are not able to do the test at 25oC as we do not have the thermo-bath.  In fact, a full-scale experimentation for batteries at different conditions and with different type of batteries will be necessary in order for this method to be adopted in industry.  This requires extensive resources and will take quite some time.  We hope to use this manuscript as our preliminary finding to secure more grant for this full-scale experimentation. These additional details are added into the manuscript and are highlighted in yellow on Page number 13.

Table 12 Verification of the model for additional three more batteries tested at 55oC and 5C discharging current.

Sample #

SoH (Using ECBE)

SoH estimation using our generalized k values

% error in SoH estimation with generalized k values

1

89.06

(After 100 cycles)

88.24

0.92

2

91.39

(After 100 cycles)

90.72

0.73

3

90.66

(After 100 cycles)

89.94

0.79

Line 115: Reproducibility is what you measure yourself! It is about your paper, not what others do.

Answer:

Thanks to the reviewer and the details are added as answered earlier.

Line 117: more batteries will be tested when resource is available. Why did you sent then this manuscript since you understand perfectly well you need reproducibility by testing more than one battery for each condition. This statement in Line 117 is actually why this paper cannot be accepted at this stage. 

Answer:

Thanks to the reviewer and the details are added as answered earlier.

Figure 1: Why is charge voltage higher at 55 degrees Celsius?

Answer:

This is an interesting question.  Figure 1 shows the typical time progression of the terminal voltage of a LiB during charging and discharging.  One can see that the voltage is higher when LiB is charging at higher temperature. This is because cell impedance will be higher at higher temperature. The terminal voltage can be modelled as the constant current multiples the cell impedance as the charging is done using CC-CV, and hence a higher terminal voltage is resulted.  In fact, the cell impedance of a LiB has been employed to monitor the temperature of LiB by Beelen et. al. [REF]

Ref: Beelen, H., Mundaragi Shivakumar, K., Raijmakers, L., Donkers, M. C. F., & Bergveld, H. J. (2020). Towards impedance‐based temperature estimation for Li‐ion battery packs. International Journal of Energy Research44(4), 2889-2908.

The details are added in manuscript on Page 4.

Figure 3: All letters are not well readable! Please make this Figure again.

Answer

Figure quality is improved.

Figure 4 is not clear to me. Explain again.

Answer:

We apologize for the confusion. It is explained and highlighted on Page 6.

To demonstrate the effect of varying C-rate at constant temperature and the effect of varying temperatures at constant C- rate on SoH estimation accuracy of the SECF model, we divide our results into 4 sections as shown in Figure 4. In Figure 4, A, B, C and D represent cases where k values are obtained from batteries tested at 1C_55oC, 3C_55oC, 1C_25oC and 3C_55oC respectively. On the other hand, A’ and B’ represent the SoH estimation errors for cells A and B using k values obtained from cell A.  C* and D* represent the SoH estimation errors for cells C and D using k values obtained from cell A.  Similarly, A’’ and B” are the SoH estimation errors for cells A and B using k values obtained from cell B.  Others follow the same notations.  

Figure 4 Notations used in the analysis of test results.

General: use in the text k1 and not k1, use k2 and not k2, use k3 and not k3.

Answer

Thank for pointing them out. The notations are modified as suggested.

Line 234-235: What is temp range lab? Why were tests not performed in a thermostatted oil bath?? Now the temperature is not defined and that is not acceptable.

Answer

Thanks for the comment.

The room temperature of the lab is controlled at 25oC, and it is fairly steady as our daily temperature chart shows a variation between 24-26oC. Unfortunately, we do not have the thermostatted oil bath in our lab.  We also realized that due to the small fluctuations in the lab temperature at 25oC, larger error was obtained as mentioned in the manuscript on Page 8. In fact, this is also close to the actual scenario, where temperature fluctuates in the normal environment, and even with such fluctuation, our estimation error is still good.

We have modified our paragraph on Page 9 as follows.

Original: In comparing the % errors for C’ and A’, and that for D’ and B’, we can see that the % error is larger for the case of 25oC. Such larger error is due to the inaccuracy of the temperature of 25oC as the testing was done in the laboratory ambient, and the ambient temperature has a small fluctuation over the span of measurement period.

Revised: In comparing the % errors for C’ and A’, and that for D’ and B’, we can see that the % error is larger for the case of 25oC. Such larger error is due to the inaccuracy of the temperature of 25oC as the testing was done in the laboratory ambient, and the ambient temperature has a small fluctuation over the span of measurement period. The temperature varies between 24 to 26oC during our test.

Line 350-351: You conclude yourself that data are too limited so it is no surprise I conclude the same. 

Answer:

Thanks for the suggestion. We have added more data.

I trust upon your expertise to:

  • run more tests with at least 3 batteries for one condition
  • stabilize the temperature in a temperature regulated oil bath
  • explain in more detail the model with k1, k2and k3 as parameters
  • have the entire text be corrected by a native English speaker
  • strongly improve the Figures and graphics to make them more readable

I will advice the Editor the following:

The topic is very interesting but the revision needed will take more time than acceptable. Hopefully you will be able to find financial resource.

This means the peer review process end here for this manuscript as you first need to run more tests. I am very pleased to review your renewed manuscript. I wish you success in performing the additional tests and to take care to of the points as mentioned by me. 

As you see, the topic is very interesting but we simply need more and better defined tests.

Kind regards,

Reviewer

Reviewer 2 Report

The presented study is of significant interest to the readership. The study is well structured and presented.

Several questions arose while working through the document. It would be great if these questions could be clarified within the document:

1) The factors k1, k2, and k3 where determined for 4 different cycling protocols, however, only with one cell each. Hence, no error on the k value is obtained. It is reported in literature that cells deviate in aging by several percent. How is this impacting the proclaimed accuarcy of 99%?

2) The cycling protocols chosen where only varied with regard to temperature and c-rate, however, also depth of discharge has a significant impact on the aging. How would the model predict SOH and remaining usable life for a cell which is only using a fraction of the capacity?

3) The authors state in the introduction that the aging is dependent on cell design, however, here only one design is tested. How can it be proven that this model is also valid for different cell designs?

3) It is known in literature that aging can also be non linear over cycle life. How would the model cope with such a behavior?

4) With regard to table 8 the authors state that the actual cell temperature is higher than the temperature in the climate chamber and increases with c-rate. which is a typical observation related to the cell impedance. However, the authors do not comment on the lower temperatures of the cell at 55 °C of the climate chamber. How is it possible that these cells have a lower temperature?

The authors themselves are regretting the minimal availability of test data, reasoned by the low amount of resources. In the conclusions it is stated that the experiments have to be repeated with more cells and also with different cell types. Here the question arises if the paper is adding to the knowledge of the community as the soundness of the results remains unclear.

Author Response

Dear Editor,

We appreciate very much the effort of your team and the comments from Reviewers.  They are very good and have indeed help to improve our manuscript significantly. We took every comments seriously, and below are our replies for your considerations.  Thank you

We have also attached the manuscript files with and without highlights.

Reviewer 2

The presented study is of significant interest to the readership. The study is well structured and presented.

Several questions arose while working through the document. It would be great if these questions could be clarified within the document:

 1) The factors k1, k2, and k3 where determined for 4 different cycling protocols, however, only with one cell each. Hence, no error on the k value is obtained. It is reported in literature that cells deviate in aging by several percent. How is this impacting the proclaimed accuracy of 99%?

 Answer:

You are absolutely correct, and thank for pointing it out. We acknowledge your comment and we have removed the claim of providing 99% accuracy from our present manuscript.  We hope to provide such claims in future as we continue on this work.

2) The cycling protocols chosen where only varied with regard to temperature and c-rate, however, also depth of discharge has a significant impact on the aging. How would the model predict SOH and remaining usable life for a cell which is only using a fraction of the capacity?

 Answer:

We thank the reviewer for such excellent question. To predict SoH and RUL based on a fraction of the capacity consumption is the strength of this model.  The reason that we can predict SoH based on a fraction of the capacity, instead of using the coulomb counting method is based on the principle that if a battery cell has lesser stored charge, its terminal voltage will drop faster for a given discharge current.  Hence, by checking the rate of decrease in the terminal voltage, and through electrochemistry theory, one can estimate the stored charge in the cell, and this is the basis of our ECBE model published in JPS in 2014.  In fact, based on the ECBE model, we will be able to track how the SoH degrades with the charge-discharge cycles.  Since our proposed model obtained its k parameters using ECBE model, thus we are able to track the degradation pattern of SoH with the cycles.  We also know that this degradation also depends on the discharge current and temperature of the cells, as it is also included in the ECBE model, so our model includes temperature term and discharge current. Once the model parameters are determined, it is natural to perform inverse calculation to determine the RUL as illustrated in the manuscript.

We have indeed overlooked the impact of the depth of discharge in our modelling, and we also did not perform experiments on the different depth of discharge.  This will be mentioned in our revised manuscript and included this as our future work.

3) The authors state in the introduction that the aging is dependent on cell design, however, here only one design is tested. How can it be proven that this model is also valid for different cell designs?

Answer:

Battery degradation is definitely depending on the cell design as cell design affect the heat dissipation and the uniformity of the temperature within the cells.  In this manuscript, we demonstrated that our model works well for this design (we also added three more batteries of the same design to further enhance the confidence of this model), however, we do not claim that it will work likewise for other design.  This need to be verified in our future work and will take a lot of time to do so.  On the other hand, based on the basic principle as stated above, and that the ECBE model is verified for LCO, LFP and NMC, we believed that our proposed model should also work, but with different set of k parameter values. 

3) It is known in literature that aging can also be non linear over cycle life. How would the model cope with such a behavior?

 Answer:

As can be seen in our paper [REF] on the ECBE model, such non linearity on the Qm, i.e. the maximum stored charge, can be modelled.  We found that larger non linearity occur in the first 50 cycles, and the subsequent aging is quite linear. Hence, when our model parameters are obtained from this ECBE model, we do not consider the first 50 cycles to avoid the high degree of non-linearity as after all, it is unlikely that the lifetime of a cell is less than 50 cycles. 

The information is added in the manuscript on Page 5 and is highlighted.

There are large fluctuations or non-linearity observed in Qm over the first few cycles, and this non-linearity is higher at lower temperatures which is also observed by other researchers [REF], thus the first 50 cycles are not used for the k values extraction.

Ref: Leng, Feng, et al. "Hierarchical degradation processes in lithium-ion batteries during ageing." Electrochimica Acta 256 (2017): 52-62.

4) With regard to table 8 the authors state that the actual cell temperature is higher than the temperature in the climate chamber and increases with c-rate. which is a typical observation related to the cell impedance. However, the authors do not comment on the lower temperatures of the cell at 55 °C of the climate chamber. How is it possible that these cells have a lower temperature?

 Answer

We appreciate the Reviewer in spotting this error. We apologize for the mistake. We have re measured the temperature rise for the batteries and the revised Table 8 for the actual cell temperature is as shown below.

Table 8 Cell temperature when discharge at different C rate under two ambient conditions

1C rate

3C rate

25oC Ambient

34.86 oC

42.32 oC

55oC Ambient

58   oC

63   oC

The authors themselves are regretting the minimal availability of test data, reasoned by the low amount of resources. In the conclusions it is stated that the experiments have to be repeated with more cells and also with different cell types. Here the question arises if the paper is adding to the knowledge of the community as the soundness of the results remains unclear.

Answer

We totally agree with the reviewer on the scarcity of the data.  We have thus added three more batteries of the same design and the results turn out to be encouraging, with small % error, as stated below.  In fact, a full-scale experimentation for batteries at different conditions and with different type of batteries will be necessary in order for this method to be adopted in industry.  This requires extensive resources and will take quite some time.  We hope to use this manuscript as our preliminary work to secure more grant for this full-scale experimentation.  The following paragraph is added:

For the purpose of reproducibility, 3 similar batteries are tested at 55oC _5C and the % error in the SoH estimation is estimated using Equations (3) and (4) which are presented in Table 12. These conditions are chosen in order to save time as the battery degradation is fastest in such harsh conditions. Batteries are tested for 100 cycles and the % error is estimated. % error in SoH estimation with generalized k values from Equation (3) and (4) clearly verifies that the method proposed in this work for other samples as well.

Table 12  Verification of the model for additional three more batteries tested at 55oC and 5C discharging current

Sample #

SoH (Using ECBE)

SoH estimation using generalized k values

% error in SoH estimation with generalized k values

1

89.06

(After 100 cycles)

88.24

0.92

2

91.39

(After 100 cycles)

90.72

0.73

3

90.66

(After 100 cycles)

89.94

0.79

Reviewer 3 Report

The authors have proposed an on-line SOH estimation for lithium ion batteries. The manuscript has novelty. It is well-organized. The topic is interesting and timely. This Reviewer likes the innovation; however, the manuscript suffers from several minor issues, which are as follows.

1) Please include problem statements and rationale in the Abstract section.

2) Please elaborate upon the merits and demerits of the proposed method from practical standpoints. For example, the implementation of such storage systems is a must in the future power systems, especially active distribution systems. In this regard, please extend the literature review section by considering a variety of applications. As an illustration, a new bi-objective approach to energy management in distribution networks with energy storage systems; a two-stage algorithm for optimal scheduling of battery energy storage systems for peak-shaving, NAPS 2019; Impact of operational decisions and size of battery energy storage systems on demand charge reduction, 2019 Milan PowerTech; optimal scheduling of battery energy storage systems for solar power smoothing, 2019 Southeast Conf.

3) Please provide an intelligible explanation on current limitation as well as future scope of extension of the work.

4) Some figures are not suitable, please double-check the quality of all figures, more specifically Fig. 3, which is very difficult to read. 

Author Response

Dear Editor,

We appreciate very much the effort of your team and the comments from Reviewers.  They are very good and have indeed help to improve our manuscript significantly. We took every comments seriously, and below are our replies for your considerations.  Thank you

We have also attached the manuscript files with highlights.

Reviewer 3

The authors have proposed an on-line SOH estimation for lithium ion batteries. The manuscript has novelty. It is well-organized. The topic is interesting and timely. This Reviewer likes the innovation; however, the manuscript suffers from several minor issues, which are as follows.

  • Please include problem statements and rationale in the Abstract section.

Answer

Thanks for the suggestion, and we modify the abstract as follows.

Inaccurate SoH estimation of battery can lead to over-discharge as the actual depth of discharge will be deeper, or more than necessary number of charging as the calculated SoC will be under-estimated, depending on the inaccuracy in the maximum stored charge is over or under estimated.  Both can lead to increase degradation of a battery. Inaccurate SoH can also lead to the continuous use of battery below 80% actual SoH that could lead to catastrophic failures.  Therefore, an accurate and rapid on-line SoH estimation method for lithium ion batteries under different operating conditions such as varying ambient temperatures and discharge rate is important.  This work develop a method for this purpose, and the method combines the electrochemistry based electrical model and semi-empirical capacity fading model on a discharge curve of a lithium-ion battery for the estimation of its maximum stored charge capacity, and thus its state of health.  The method developed produces a close form that relate SoH with the number of charge-discharge cycles as well as operating temperatures and currents,  and its inverse application allows us to estimate the remaining useful life of LiB for a given SoH threshold level.  

  • Please elaborate upon the merits and demerits of the proposed method from practical standpoints. For example, the implementation of such storage systems is a must in the future power systems, especially active distribution systems. In this regard, please extend the literature review section by considering a variety of applications. As an illustration, a new bi-objective approach to energy management in distribution networks with energy storage systems; a two-stage algorithm for optimal scheduling of battery energy storage systems for peak-shaving, NAPS 2019; Impact of operational decisions and size of battery energy storage systems on demand charge reduction, 2019 Milan PowerTech; optimal scheduling of battery energy storage systems for solar power smoothing, 2019 Southeast Conf.

Answer

Thanks reviewer for the interesting suggestion. We have extended the discussion on the practicality of our proposed method from EVs to Energy storage where the accurate estimation of the SoH and Remaining useful life (RUL) are important. The information is added into the manuscript on Page 2.

Accurate estimations of SoH and RUL are also crucial in energy storage applications.  As the charging and discharging tend to be more often when LiB is used for energy storage, especially under solar and wind energy systems, large RUL with respect to charging and discharging will be important for a specific SoH threshold to justify the system cost.  A sufficiently high SoH of LiB should be maintained for energy storage so that it can provide enough energy for usage when the solar energy or wind energy is no longer available.  This model allows such estimation of the RUL.

Along with the merit, we have also mentioned the limitations of the proposed method in our work which drive us to do further research on Page 14 and is highlighted here.

As mentioned in the introduction, the proposed model does not include the effect of the depth of discharge which is known to affect SoH.  This is one of the limitation of the proposed model and the inclusion of the depth of discharge in the model will also be our future work. Another limitation is the possible change of the values of the k parameters when the SoH becomes low as the internal structure of the LiB could have changed so significantly that a new set of k parameters are to be extracted from the ECBE model which has shown to be accurate even at low SoH [my JPS paper]. Such investigation will also be our future work.

      Ref: Leng, Feng, et al. "A practical framework of electrical based online state-of-charge estimation  

      of lithium ion batteries." Journal of Power Sources 255 (2014): 423-430.

  • Please provide an intelligible explanation on current limitation as well as future scope of extension of the work.

Answer

We are not clear the meaning of “current limitation” as it could mean the discharge current or it means present limitation.

We have demonstrated that our model is accurate even up to 5C discharge rate, although discharge beyond 3C is dangerous.

The present limitation of the model is the lack of sufficient battery test data at more discharge rates, changing rates, temperature changes for batteries with different designs and brands.  Such limitation arises because our model is not a first principal model, but it is semi-empirical.  All the above mentioned will constitute our future work, and we hope that this manuscript can form a preliminary work for us to secure a larger grant to perform the future work.  We also hope to have this manuscript to initiate collaborative works with other research labs to speed up the verification of this model to a larger database as we see that this will be helpful for the industry.

4) Some figures are not suitable, please double-check the quality of all figures, more specifically Fig. 3, which is very difficult to read. 

Answer

All Figure qualities are improved in the manuscript.

Round 2

Reviewer 1 Report

Dear authors,

I am very pleased to read your revised document.

You have given a good answer to the questions raised and I therefore accept this revised paper.

Congratulations with this important work.

Kind regards,

Reviewer

Reviewer 2 Report

All questions have been adressed.

A minimum estimation of error was adressed by adding some addtionial data for at least one testing protocol.

The small amount of tests still limits the potential of the paper.

Reviewer 3 Report

Thanks for addressing the comments.